# A Universal LC-MS/MS Method for Simultaneous Detection of Antibiotic Residues in Animal and Environmental Samples

**DOI:** 10.3390/antibiotics11070845

**Published:** 2022-06-24

**Authors:** Chak-Lun Chan, Hogan Kok-Fung Wai, Peng Wu, Siu-Wai Lai, Olivia Sinn-Kay Chan, Hein M. Tun

**Affiliations:** 1School of Public Health, Li Ka Shing Faculty of Medicine, The University of Hong Kong, Hong Kong SAR, China; darrencl@connect.hku.hk (C.-L.C.); hkfwai@connect.hku.hk (H.K.-F.W.); pengwu@hku.hk (P.W.); swlai@hku.hk (S.-W.L.); oliviachan@hku.hk (O.S.-K.C.); 2HKU-Pasteur Research Pole, School of Public Health, Li Ka Shing Faculty of Medicine, The University of Hong Kong, Hong Kong SAR, China; 3The Jockey Club School of Public Health and Primary Care, The Chinese University of Hong Kong, Hong Kong SAR, China; 4Microbiota I-Center (MagIC), The Chinese University of Hong Kong, Hong Kong SAR, China

**Keywords:** antibiotic residue, solid-phase extraction, LC-MS/MS

## Abstract

Detecting and monitoring the usage of antibiotics is a critical aspect of efforts to combat antimicrobial resistance. Antibiotic residue testing with existing LC-MS/MS methods is limited in detection range. Current methods also lack the capacity to detect multiple antibiotic residues in different samples simultaneously. In this study, we demonstrate a methodology that permits simultaneous extraction and detection of antibiotic residues in animal and environmental samples. A total of 30 different antibiotics from 13 classes could be qualitatively detected with our methodology. Further study to reduce analytes’ matrix effect would allow for quantification of antibiotic residues.

## 1. Introduction

Antibiotics have been in use for nearly a century and have been an important means to treat and prevent bacterial infection in both humans and animals [1]. However, the misuse and overuse of antibiotics have driven the rapid development of drug-resistant bacteria [2]. This has contributed to the bigger problem of antimicrobial resistance (AMR), which is currently a severe global public health issue. In order to tackle the issue of AMR, a One Health holistic approach, which covers human, animal, and environmental sectors, is necessary, due to inter-sectoral transmission [3].

Under the One Health framework, governments and organizations have taken a primary approach to mitigating AMR by reducing antimicrobial use (AMU) in human and animal sectors [4,5]. However, AMU and antimicrobial consumption have been difficult to measure. AMU survey or antimicrobial procurement data serve as a proxy to measure AMU but are limited by data availability and reliability. Thus, testing of antibiotic residues would be a complementary or alternative option.

In antibiotic residue testing, most of the fundamental laboratory detection methods involve an initial extraction followed by liquid chromatography tandem mass spectrometry (LC-MS/MS) (Figure 1). Although the fine details of the methodology differ between different studies, this fundamental approach has been used to detect the chemical composition of and antibiotic residues in different samples, including porcine muscle [6], duck meat [7], aquaculture products [8], bovine milk [9], milk [10], honey [11], natural water [12,13], swine manure [14], and distiller grains [15]. 

However, the common limitations of antibiotic residue testing in these studies are the lack of proven applicability of the testing protocol to detect different sample types and the wide number of antibiotic classes used in human and animals. For example, effectively extracting target chemicals from solid samples is different from that of water samples, and the classes of antibiotic that can be detected vary. Moreover, antibiotics and their residues can be unstable. For example, these substances may be unstable in water [16] and could be unstable based on their storage conditions [17,18]. To address these shortcomings, we demonstrate how a single extraction and detection method can be used in AMU surveillance to simultaneously detect multiple antibiotics in both animal and environmental samples.

## 2. Materials and Methods

### 2.1. Chemicals and Reagents

Acetonitrile, ammonia solution, citric acid monohydrate, dibasic sodium hydrogen phosphate, and LC-MS grade methanol were purchased from VWR (Radnor, PA, USA) for chemical extraction. The methanol that was used as the solvent in LC-MS was procured from Wako Chemicals (Osaka, Japan). Ethylenediaminetetraacetic acid disodium salt (Na_2_EDTA) was purchased from BDH (Radnor, PA, USA), while formic acid was purchased from Fisher Scientific (Waltham, MA, USA). A total of 43 antibiotics belonging to 16 different antibiotic classes were used for this study (Table 1). Amoxicillin, ceftazidime, cefuroxime sodium salt, chloramphenicol, ciprofloxacin hydrochloride, colistin sulfate, doxycycline monohydrate, levofloxacin, florfenicol, metronidazole, oxytetracycline, spectinomycin hydrochloride pentahydrate, and sulfadiazine were purchased from Abcam (Cambridge, UK). Mequindox was ordered from Huawen Chemical (Henan, China), while tylvalosin was purchased from Santa Cruz Biotechnology (Dallas, TX, USA). Erythromycin, gentamicin sulfate, tylosin tartrate salt, and trimethoprim were purchased from MP Biomedicals (Irvine, CA, USA). Ampicillin, caffeine-(trimethyl-^13^C_3_) solution (used as internal standard), cefalexin, cefquinome sulfate, ceftiofur sodium, chlortetracycline hydrochloride, clindamycin phosphate, enrofloxacin, kanamycin sulfate, lincomycin hydrochloride, meropenem, neomycin trisulfate salt hydrate, norfloxacin, ofloxacin, penicillin G sodium salt, streptomycin sulfate salt, sulfachloropyridazine, sulfadimidine, sulfamethoxazole, sulfamonomethoxine, tetracycline, tiamulin, tilmicosin, and vancomycin were purchased from Sigma-Aldrich (Darmstadt, Germany).

### 2.2. Sample Preparation

Swine feces were collected from local farms and freeze-dried before processing. Pork was purchased from local markets. River water, animal drinking water, and Milli-Q water were used as water samples. An antibiotic mixture (~0.01 mg/mL concentration) containing 43 different antibiotics was used for the spike-in. For solid samples, the antibiotic mixture was directly spiked into 1 g of solid sample (i.e., fecal sample and meat sample) by adding the aqueous antibiotic mixture into a Falcon tube containing the solid sample and then mixing by vortex. The sample was allowed to stand for 1 h to allow for the antibiotic mixture to be absorbed into the solid sample. For liquid samples, the antibiotic mixture was spiked into 100 mL of liquid sample at three volumes: 1 mL of 0.01 mg/mL (i.e., 10 μg), 0.5 mL of 0.01 mg/mL (i.e., 5 μg), and 0.1 mL of 0.01 mg/mL (i.e., 1 μg). All samples were stored at −20 °C and extracted within 1 week.

### 2.3. Chemical Extraction

Chemical extraction was carried out according to the literature [14] with modifications. The chemical extraction buffer that was used was a mixture of Na_2_EDTA-McIlvaine buffer solution (10 mL), which contains 10.93 mg/mL anhydrous dibasic sodium phosphate, 12.93 mg/mL citric acid monohydrate, and 37.22 mg/mL Na_2_EDTA, and 100% methanol (10 mL). For extraction, 20 mL of this chemical extraction buffer was added into 1 g of sample (fecal sample or meat sample). Extraction was carried out in a shaking incubator at 300 rpm for 30 min at room temperature. Supernatants were collected after centrifugation at 3200 g for 5 min at 4 °C. Residues were further extracted two more times with the same protocol. Supernatants of the same sample were then combined and stored at −20 °C until solid-phase extraction (SPE) was performed. For liquid samples, chemical extraction was not performed. All samples were filtered through a 0.2 μm PES syringe filter, purchased from Sartorius (Göttingen, Germany), prior to solid-phase extraction.

### 2.4. Solid-Phase Extraction (SPE)

An Oasis MAX 6cc (150 mg) cartridge (cat. no.: 186000369), Oasis PRiME HLB 6cc (200 mg) cartridge (cat. no.: 186008057), and Oasis PRiME MCX 6cc (150 mg) cartridge (cat. no.: 186008919) were purchased from Waters (Milford, MA, USA) and used to form a MAX-HLB-MCX combined cartridge for solid-phase extraction. All cartridges were pre-conditioned with acetonitrile and Milli-Q water separately. To investigate the retention of antibiotics in different SPE cartridges, 10 mL of the antibiotic mixture containing 43 targeted antibiotics was loaded to the MAX-HLB-MCX combined cartridge. For solid samples, supernatants collected from chemical extractions were diluted with Milli-Q water to reduce the methanol content to less than 5% (*v*/*v*). All of the diluted supernatant flowed through the MAX-HLB-MCX combined cartridge at a flow rate of 3 mL/min. For liquid samples, all of the samples were passed through the MAX-HLB-MCX combined cartridge directly at a flow rate of 3 mL/min. Analytes were eluted from the MAX cartridge, HLB cartridge, and MCX cartridge separately. To elute the analytes, 4 mL of 2% formic acid in methanol, acetonitrile/methanol (60%/40%; *v*/*v*), and 5% ammonia solution in methanol were used, respectively. The elution process was repeated three times. The ratio in which elutes were mixed from the MAX, HLB, and MCX cartridges was 1:1:1. Elutes from the same sample were combined for LC-MS/MS analysis.

### 2.5. LC-MS/MS Analysis

The Acquity I-Class ultra-high-performance liquid chromatographic system by Waters (Milford, MA, USA), coupled with the QTRAP^®^ 6500+ MS system from AB Sciex (Framingham, MA, USA), was used for LC-MS/MS analysis. A Phenomenex Synergi 4 μm Fusion-RP 80 Å (2 mm × 50 mm) column was used for separation, and the column oven temperature was set at 40 °C. The elution gradient (solvent A: 0.1% aqueous formic acid, solvent B: acetonitrile) was set up as follows: 0 min, 0% B; 0.1 min, 0% B; 1 min, 10% B; 6.5 min, 50% B; 7 min, 100% B; 8 min, 100% B. This gradient was re-equilibrated to 0% B for 2 min after each run. The flow rate was 0.5 mL/min, and the injection volume was 5 μL. In terms of electrospray ionization, the parameters were the following: (CUR), nitrogen, 12; collision gas (CAD), high; electrospray voltage, +5500 V; ion source temperature, 550 °C; curtain gas of 25, CAD gas medium, and gas 1 and 2 of 45 and 50 psi, respectively. Retention time and transitions are shown in Table 2.

For transition ranges, each pure antibiotic compound purchased commercially was first injected into the LC-MS/MS instrument for preliminary testing. From this preliminary test, information about transition ranges was obtained. Two transitions of each antibiotic with sharp peaks shown were chosen as references to identify the antibiotic.

### 2.6. Data Analysis

Data analysis for LC-MS/MS was performed using the SciEX OS-Q Analysis Software (Framingham, MA, USA). Analytes were confirmed by comparing the retention time and the ratio of characteristic transitions between the sample and the standard.

## 3. Results

### 3.1. Limit of Detection

This study began with the determination of the detection limit (i.e., limit of detection, LOD) of the MS system. The LOD was determined by injecting a low concentration of antibiotic standard into the mass spectrometer directly and then reviewing the peak generated. If the signal was three times higher than the background base noise level, then we accepted the peak as an actual peak. Infusion was only used for optimizing the MRM parameter before LC-MS/MS. The results of LOD are shown in Table 2. Generally, the detection limit varied. Most of the 43 antibiotics were detectable at levels lower than 70 ppb, with only six antibiotics having a level of detection higher than 70 ppb. These six antibiotics were colistin A, colistin B, neomycin trisulfate, gentamicin, and streptomycin sulfate. Nevertheless, all 43 targeted antibiotics were detectable.

### 3.2. Solid-Phase Extraction

The percentage of recovery was calculated by comparing the concentration of antibiotic recovered after passing through SPE and without passing through SPE. In other words, the percentage of recovery = total concentration of antibiotic in elutes from SPE/concentration of antibiotic in antibiotic mixture before passing through SPE. The recoveries of amoxicillin, ampicillin, cefquinome sulfate, meropenem, and tiamulin from SPE were poor, with loss being >75% (Table 3). Apart from those which had a poor recovery, 11 antibiotics were retained in the MAX cartridge; five antibiotics were retained in the HLB cartridge; and 17 antibiotics were retained in the MCX cartridge. For chlortetracycline hydrochloride, doxycycline, and mequindox, the MAX cartridge could not completely retain all of the residues, and a significant portion flowed through the MAX cartridge and were retained in the HLB cartridge. For sulfadiazine and sulfadimidine, they could be detected in the elutes of all three cartridges in the MAX-HLB-MCX tandem. Overall, 38 out of the 43 antibiotics had a recovery that was satisfactory or good after SPE.

### 3.3. Chemical Extraction

The integrated recovery of antibiotic residues (i.e., including the limitation of chemical extraction, SPE, and LOD) is shown in Table 4. Using our methodology, 30 out of 43 targeted antibiotics could be detected (Table 4). The 13 antibiotics that could not be detected were: amoxicillin, cefquinome sulfate, ceftazidime, cefuroxime, ciprofloxacin, colistin (A and B), gentamicin, kanamycin sulfate, meropenem, neomycin trisulfate, norfloxacin, and vancomycin. Among these 13 undetectable antibiotics, amoxicillin, cefquinome sulfate, and meropenem were found to have a poor recovery from SPE.

### 3.4. Sensitivity

The sensitivities of antibiotic classes were calculated by taking the raw data and then summing the data for all of the tests performed for all antibiotics in a class. Overall, the sensitivity of the protocol in this study was high (i.e., >60%) for 30 out of the 43 antibiotic residues (Table 5). When the spike-in concentration was high, i.e., 10 μg, our approach showed a sensitivity of 100% in most of the antibiotics in all three types of sample. For ampicillin, ceftiofur, chloramphenicol, chlortetracycline, doxycycline, erythromycin, metronidazole, penicillin, spectinomycin, streptomycin, tetracycline, tilmicosin, tylosin tartrate, and tylvalosin, the sensitivities of the protocol were only reduced when the spiked-in antibiotic content was reduced to 1 μg (Appendix A).

Out of 16 classes of antibiotic tested, antibiotics from 13 classes could be detected (Table 6). The sensitivity of detecting antibiotics from the antifolate, lincosamide, pleuromutilin, quinoxaline 1,4-di-N-oxide (QdNO), and sulfonamide classes was relatively high in all three types of sample at three different concentrations. The sensitivity of detection of antibiotics from the amphenicol, macrolide, nitroimidazole, and tetracycline classes was high (around 100%) when the spiked-in content was 10 μg and 5 μg, but it was less sensitive when the spiked-in content was reduced to 1 μg. The sensitivity of our detection method was relatively lower for aminoglycosides, cephalosporins, fluoroquinolones, and penicillins, while carbapenems, glycopeptides, and polymyxins could not be detected.

## 4. Discussion

In this study, we have developed a LC-MS/MS-based working protocol that could detect residues of 30 antibiotics from 13 classes in animal meat and environmental samples. Although using LC-MS/MS to detect antibiotic residues in food and water samples is not a novel technique, our protocol has two significant improvements. The first improvement is that we have developed a methodology that can cover different sample types. Previous developed methods can detect 34 veterinary drugs from six distinct groups in porcine muscle [6]; 75 antibiotics from six groups in meat and aquaculture products [8]; 63 pharmaceuticals in natural water [12]; 58 antibiotics from eight groups in milk [10]; and 20 antibiotics from three different groups in honey [11]. Most of these methods targeted food samples and are not sufficient in monitoring antibiotic contaminations, particularly in various types of environmental sample.

The second improvement is that we have developed a methodology that can cover a wider range of different antibiotics. Different antibiotics have different chemical properties, and they may require different extraction and detection methods [19,20]. In our study, we used a single extraction and detection protocol to cover 30 antibiotics from 13 families, which makes it an improvement over existing methodologies in terms of efficiency and convenience [7,9,13,14,15]. To provide an insight into why we could make the aforementioned two improvements in detecting antibiotic residues, we provide an interpretation of the results after SPE and chemical extraction were conducted in our protocol.

In theory, SPE is a common procedure performed to clean up the samples before LC-MS/MS analysis [21,22]. However, the major purpose of using SPE in this study was to concentrate the analytes from large-volume but low-concentration samples, such as environmental water and wastewater with diluted concentrations of antibiotic residues. Thus, in order to increase the sensitivity of detection, SPE was an essential step of our developed method. The chemical nature of the antibiotics and the type of SPE cartridge could influence the final recovery of antibiotic residues for detection. Considering that 43 antibiotics with varying properties were targeted in this study, we used a MAX-HLB-MCX tandem consisting of the MAX cartridge, the MCX cartridge, and the HLB cartridge to retain as much of the antibiotic residues as possible. The rationale behind this tandem formation was that the different cartridges would target compounds with different pH properties: the MAX cartridge targets acidic compounds; the MCX cartridge targets basic compounds; and the HLB cartridge targets relatively neutral compounds (i.e., those that are neither basic nor acidic).

Our SPE method was able to recover a majority of the antibiotics (38 out of 43 antibiotics) with satisfactory or good performance, indicating its ability to successfully detect antibiotics of different pH properties. For those five antibiotics that had a poor recovery after SPE (i.e., >75% loss), we hypothesized that the chemical structure of these antibiotics may not be compatible with the tandem formation that we designed. Most of these belonged to the β-lactam class of antibiotics, of which amoxicillin, ampicillin, cefquinome sulfate, and meropenem possess a chemically unstable β-lactam ring that spontaneously undergoes hydrolysis [23]. Thus, this is a plausible reason explaining why it was very difficult to recover these antibiotics in water samples.

Generally, the extraction method of using Na_2_EDTA-McIlvaine buffer solution/methanol (1:1; *v*/*v*) provided a detection of 30 different antibiotics. There were three antibiotics that were found to have a poor SPE recovery and could not be detected in spiked-in samples: amoxicillin, cefquinome sulfate, and meropenem. It would not be possible to determine the reason of zero sensitivity for these three antibiotics, since it could be related to the chemical extraction method, poor SPE recovery, or both. Thus, it is difficult to determine whether the three antibiotics could be extracted using our chemical extraction protocol. For the rest of the nine antibiotics that could not be detected, it seems that the extraction method was insufficient in recovering these antibiotics. We arrived at this conclusion because we observed a satisfactory recovery rate from SPE for these antibiotics, and the spike-in concentrations were higher than the LOD. Based on these results, the extraction method may not be suitable for extracting the residues of ceftazidime, cefuroxime, ciprofloxacin, colistin, gentamicin, kanamycin sulfate, neomycin trisulfate, norfloxacin, and vancomycin from samples. Further study for modifying the chemical extraction method to further increase the range of recovery may be necessary.

The proposed method has notable limitations, such as the inability to conduct precise quantitative analysis and a decline in sensitivity with lowering the spiked-in concentration. Although SPE is commonly performed to remove impurities from analytes in order to reduce a potential matrix effect, the use of the MAX-HLB-MCX tandem resulted in strong matrix effects. Moreover, it is difficult to obtain corresponding isotopes for all tested antibiotics to correct the matrix effect of each antibiotic during quantification. However, an estimation of the antibiotic concentrations may still be made by performing calculations. From LC-MS/MS results, one will have the information of the total amount of antibiotics in the elute of each sample. The concentration of an antibiotic can be estimated by the following equation: (concentration of antibiotic in sample = amount of antibiotic/sample weight or sample volume). Thus, the difference in volume between solid and liquid samples during the whole process does not affect the estimation of the antibiotic concentration in samples after calculation.

Only 15 out of the 30 detectable antibiotics could maintain a sensitivity of 100% when the spiked-in content was reduced from 10 μg to 5 μg and 1 μg for all three types of sample. It seems that, when the spiked-in concentrations were gradually lowered, the sensitivity of the protocol decreased. From the results in Table 3 and Table 4, the loss of antibiotic residues during the extraction process and SPE was expected. The explanation is that the likelihood of a false negative result increased when the residue content decreased. Having stated these limitations, our method could be useful in AMU surveillance in livestock farms as a first-line qualitative assessment tool—especially for detecting residues in farm waste.

## 5. Conclusions

In conclusion, 30 different antibiotics from 13 classes could be detected with high sensitivity with our sample processing method when the residue content was 10 ppm or above. When the residue content was reduced to 1 ppm, 27 different antibiotics could still be detected, and 21 of them had a sensitivity higher than 50%. The developed chemical extraction method, together with SPE, allowed us to detect at least 30 antibiotic residues from 13 families qualitatively in foods and environmental samples at the same time. Nevertheless, further study to reduce the matrix effect of analytes is necessary so that quantification of antibiotic residues could be possible.

## Figures and Tables

**Figure 1 antibiotics-11-00845-f001:**
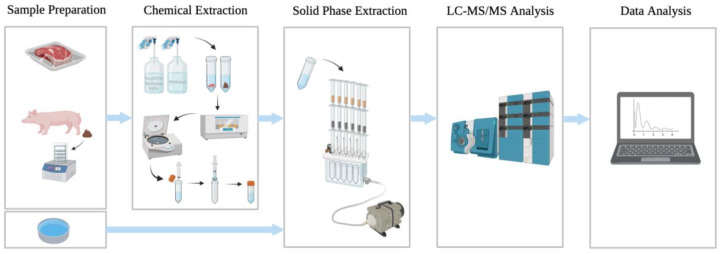
Flowchart showing the key steps in antibiotic residue testing. The figure was created with BioRender.com.

**Table 1 antibiotics-11-00845-t001:** List of antibiotics tested.

Classes	Antibiotics
Aminoglycosides	Gentamicin
	Kanamycin sulfate
	Neomycin trisulfate salt hydrate
	Spectinomycin hydrochloride pentahydrate
	Streptomycin sulfate salt
Amphenicols	Chloramphenicol
	Florfenicol
Antifolate	Trimethoprim
Carbapenems	Meropenem
Cephalosporins	Cefalexin
	Cefquinome sulfate
	Ceftazidime
	Ceftiofur sodium
	Cefuroxime
Fluoroquinolones	Ciprofloxacin
	Enrofloxacin
	Levofloxacin
	Norfloxacin
	Ofloxacin
Glycopeptides	Vancomycin
Lincosamides	Clindamycin phosphate
	Lincomycin hydrochloride
Macrolides	Erythromycin
	Tilmicosin
	Tylosin tartrate salt
	Tylvalosin
Nitroimidazole	Metronidazole
Penicillins	Amoxicillin
	Ampicillin
	Penicillin G sodium salt
Pleuromutilins	Tiamulin
Polymyxins	Colistin A
	Colistin B
Quinoxaline 1,4-di-N-oxides (QdNOs)	Mequindox
Sulfonamides	Sulfachloropyridazine
	Sulfadiazine
	Sulfadimidine
	Sulfamethoxazole
	Sulfamonomethoxine
Tetracyclines	Chlortetracycline hydrochloride
	Doxycycline
	Oxytetracycline
	Tetracycline

**Table 2 antibiotics-11-00845-t002:** Liquid chromatography and mass spectrometry of 43 antibiotics.

Antibiotics	Retention Time (min)	Transition 1 (m/z)	Transition 2 (m/z)	Limit of Detection (ppb)
Amoxicillin	0.83	366.1 > 348.9	366.1 > 208	8.51
Ampicillin	1.78	350 > 191.9	350 > 160	0.49
Cefalexin	1.79	348 > 158	348 > 174	0.9
Cefquinome sulfate	2.33	529 > 396	529 > 134	1.11
Ceftazidime	2.14	547.1 > 467.8	547.1 > 396	3.44
Ceftiofur sodium	3.53	524 > 241	524 > 285	0.31
Cefuroxime	2.57	447 > 385.7	447 > 342	4.1
Chloramphenicol	2.56	323.1 > 274.9	323.1 > 304.8	5.41
Chlortetracycline hydrochloride	2.66	479 > 444	479 > 462	4.06
Ciprofloxacin	2.35	332.1 > 313.9	332.1 > 231.1	0.4
Clindamycin phosphate	3.16	505.1 > 457	505.1 > 487.1	0.55
Colistin A	1.95	585.6 > 535.5	585.6 > 576.4	862.53
Colistin B	1.77	578.5 > 528.4	578.5 > 569.5	793.3
Doxycycline	2.94	445.1 > 428	445.1 > 267	0.49
Enrofloxacin	2.58	360 > 316	360 > 245	0.4
Erythromycin	3.83	734.3 > 576.3	734.3 > 157.9	14.52
Florfenicol	2.26	358 > 340	358 > 241	13.21
Gentamicin	2.79	500.1 > 456	500.1 > 227.1	320
Kanamycin sulfate	0.27	485 > 324	485 > 163	6.54
Levofloxacin	2.36	362.1 > 318.2	362.1 > 261.1	0.47
Lincomycin hydrochloride	1.43	407 > 126	407 > 359	0.45
Mequindox	2.18	219 > 143	219 > 185	1.7
Meropenem	1.79	384.1 > 340.1	384.1 > 297.7	1.76
Metronidazole	1.06	172 > 128.2	172 > 82.1	0.46
Neomycin trisulfate salt hydrate	0.26	615 > 293	615 > 161	345.18
Norfloxacin	2.3	320 > 302	320 > 231.2	0.54
Ofloxacin	2.35	362 > 318	362 > 261	0.43
Oxytetracycline	2.02	461 > 426	461 > 444	1.98
Penicillin G sodium salt	2.36	335 > 160	335 > 176	10.79
Spectinomycin hydrochloride pentahydrate	0.3	333 > 189	333 > 140	2.52
Streptomycin sulfate salt	4.13	582 > 174	582 > 156	425.98
Sulfachloropyridazine	2.3	285 > 156	285 > 108	0.6
Sulfadiazine	1.45	251 > 156	251 > 92	0.45
Sulfadimidine	2.08	279 > 186	279 > 156	0.36
Sulfamethoxazole	2.4	254.1 > 155.8	254.1 > 108.2	0.43
Sulfamonomethoxine	2.38	281 > 156	281 > 126	0.62
Tetracycline	2.14	445 > 410	445 > 269	0.49
Tiamulin	4.12	494 > 192	494 > 119	0.59
Tilmicosin	3.38	869.4 > 696	869.4 > 174	5.73
Trimethoprim	1.92	291.1 > 230	291.1 > 260.9	0.41
Tylosin tartrate salt	4.13	916.3 > 772	916.3 > 174	11.63
Tylvalosin	5.2	1042.3 > 814	1042.3 > 174	64.27
Vancomycin	1.95	726 > 144	725 > 144	26.63

**Table 3 antibiotics-11-00845-t003:** Solid-phase extraction of 43 antibiotics mixture.

Antibiotics	SPE Recovery	Mainly Retained
Amoxicillin	Poor	MAX
Ampicillin	Poor	MAX
Cefalexin	Good	MCX
Cefquinome sulfate	Poor	MAX/HLB
Ceftazidime	Good	MAX
Ceftiofur sodium	Good	MAX
Cefuroxime	Good	MAX
Chloramphenicol	Good	MAX
Chlortetracycline hydrochloride	Satisfactory	MAX/HLB
Ciprofloxacin	Good	MCX
Clindamycin phosphate	Good	MAX
Colistin A	Satisfactory	MCX
Colistin B	Satisfactory	MCX
Doxycycline	Satisfactory	MAX/HLB
Enrofloxacin	Satisfactory	MCX
Erythromycin	Good	HLB
Florfenicol	Good	MAX
Gentamicin	Satisfactory	MAX
Kanamycin sulfate	Good	MCX
Levofloxacin	Good	MCX
Lincomycin hydrochloride	Good	MCX
Mequindox	Satisfactory	MAX/HLB
Meropenem	Poor	MAX/HLB
Metronidazole	Good	MCX
Neomycin trisulfate salt hydrate	Good	MCX
Norfloxacin	Good	MCX
Ofloxacin	Good	MCX
Oxytetracycline	Satisfactory	MCX
Penicillin G sodium salt	Good	MAX
Spectinomycin hydrochloride pentahydrate	Satisfactory	MCX
Streptomycin sulfate salt	Good	HLB
Sulfachloropyridazine	Good	MAX
Sulfadiazine	Good	MAX/HLB/MCX
Sulfadimidine	Good	MAX/HLB/MCX
Sulfamethoxazole	Good	MAX
Sulfamonomethoxine	Good	MAX
Tetracycline	Good	MCX
Tiamulin	Poor	MCX
Tilmicosin	Satisfactory	HLB
Trimethoprim	Good	MCX
Tylosin tartrate salt	Good	HLB
Tylvalosin	Good	HLB
Vancomycin	Good	MCX

Concentration of antibiotics mixture: 10 mL, 0.01 mg/mL. Recovery < 25% is considered to be “Poor”; recovery ≥25% and ≤60% is considered to be “Satisfactory”; and recovery >60% is considered to be “Good”.

**Table 4 antibiotics-11-00845-t004:** Detection of antibiotic mixture for spike-in of water, fecal, and meat sample.

Antibiotics	Water Sample	Fecal Sample	Meat Sample
Amoxicillin	N.D.	N.D.	N.D.
Ampicillin	Detected	Detected	Detected
Cefalexin	Detected	Detected	Detected
Cefquinome sulfate	N.D.	N.D.	N.D.
Ceftazidime	N.D.	N.D.	N.D.
Ceftiofur sodium	Detected	Detected	Detected
Cefuroxime	N.D.	N.D.	N.D.
Chloramphenicol	Detected	Detected	Detected
Chlortetracycline hydrochloride	Detected	Detected	Detected
Ciprofloxacin	N.D.	N.D.	N.D.
Clindamycin phosphate	Detected	Detected	Detected
Colistin A	N.D.	N.D.	N.D.
Colistin B	N.D.	N.D.	N.D.
Doxycycline	Detected	Detected	Detected
Enrofloxacin	Detected	Detected	Detected
Erythromycin	Detected	Detected	Detected
Florfenicol	Detected	Detected	Detected
Gentamicin	N.D.	N.D.	N.D.
Kanamycin sulfate mixture of kanamycin A (main component) and kanamycin B and C	N.D.	N.D.	N.D.
Levofloxacin	Detected	Detected	Detected
Lincomycin hydrochloride	Detected	Detected	Detected
Mequindox	Detected	Detected	Detected
Meropenem	N.D.	N.D.	N.D.
Metronidazole	Detected	Detected	Detected
Neomycin trisulfate salt hydrate	N.D.	N.D.	N.D.
Norfloxacin	N.D.	N.D.	N.D.
Ofloxacin	Detected	Detected	Detected
Oxytetracycline	Detected	Detected	Detected
Penicillin G sodium salt	Detected	Detected	Detected
Spectinomycin hydrochloride pentahydrate	Detected	Detected	Detected
Streptomycin sulfate salt	Detected	Detected	Detected
Sulfachloropyridazine	Detected	Detected	Detected
Sulfadiazine	Detected	Detected	Detected
Sulfadimidine	Detected	Detected	Detected
Sulfamethoxazole	Detected	Detected	Detected
Sulfamonomethoxine	Detected	Detected	Detected
Tetracycline	Detected	Detected	Detected
Tiamulin	Detected	Detected	Detected
Tilmicosin	Detected	Detected	Detected
Trimethoprim	Detected	Detected	Detected
Tylosin tartrate salt	Detected	Detected	Detected
Tylvalosin	Detected	Detected	Detected
Vancomycin	N.D.	N.D.	N.D.

1 mL of antibiotic mixture (0.01 mg/mL) was spiked into 1 g of solid sample or 100 mL of liquid sample. N.D. represents “not detected”.

**Table 5 antibiotics-11-00845-t005:** Sensitivity of overall protocol for 43 antibiotics at spike-in concentrations of 10 μg, 5 μg, and 1 μg.

Antibiotics	Sensitivity								
	10 μg Spiked-in			5 μg Spiked-in			1 μg Spiked-in		
	Water Sample	Fecal Sample	Meat Sample	Water Sample	Fecal Sample	Meat Sample	Water Sample	Fecal Sample	Meat Sample
Amoxicillin	0	0	0	0	0	0	0	0	0
Ampicillin	100	100	100	100	100	100	66.7	0	33.3
Cefalexin	100	100	100	100	100	100	100	100	100
Cefquinome sulfate	0	0	0	0	0	0	0	0	0
Ceftazidime	0	0	0	0	0	0	0	0	0
Ceftiofur sodium	100	100	100	100	100	100	66.7	66.7	0
Cefuroxime	0	0	0	0	0	0	0	0	0
Chloramphenicol	100	100	100	100	100	100	0	0	66.7
Chlortetracycline hydrochloride	100	100	100	100	100	100	100	100	66.7
Ciprofloxacin	0	0	0	0	0	0	0	0	0
Clindamycin phosphate	100	100	100	100	100	100	100	100	100
Colistin A	0	0	0	0	0	0	0	0	0
Colistin B	0	0	0	0	0	0	0	0	0
Doxycycline	100	100	100	100	100	100	0	33.3	0
Enrofloxacin	100	100	100	100	100	100	100	100	100
Erythromycin	100	100	100	100	33.3	100	33.3	0	33.3
Florfenicol	100	100	100	100	100	100	100	100	100
Gentamicin	0	0	0	0	0	0	0	0	0
Kanamycin sulfate	0	0	0	0	0	0	0	0	0
Levofloxacin	100	100	100	100	100	100	100	100	100
Lincomycin hydrochloride	100	100	100	100	100	100	100	100	100
Mequindox	100	100	100	100	100	100	100	100	100
Meropenem	0	0	0	0	0	0	0	0	0
Metronidazole	100	100	100	100	100	100	100	33.3	66.7
Neomycin trisulfate salt hydrate	0	0	0	0	0	0	0	0	0
Norfloxacin	0	0	0	0	0	0	0	0	0
Ofloxacin	100	100	100	100	100	100	100	100	100
Oxytetracycline	100	100	100	100	100	100	100	100	100
Penicillin G sodium salt	100	100	100	100	33.3	100	0	0	0
Spectinomycin hydrochloride pentahydrate	100	66.7	100	66.7	66.7	100	0	0	0
Streptomycin sulfate salt	66.7	100	100	33.3	0	66.7	0	0	0
Sulfachloropyridazine	100	100	100	100	100	100	100	100	100
Sulfadiazine	100	100	100	100	100	100	100	100	100
Sulfadimidine	100	100	100	100	100	100	100	100	100
Sulfamethoxazole	100	100	100	100	100	100	100	100	100
Sulfamonomethoxine	100	100	66.7	100	100	100	100	100	100
Tetracycline	100	100	100	100	100	100	100	100	66.7
Tiamulin	100	100	100	100	100	100	100	100	100
Tilmicosin	66.7	100	100	66.7	100	100	0	33.3	33.3
Trimethoprim	100	100	100	100	100	100	100	100	100
Tylosin tartrate salt	100	100	100	100	100	100	100	100	66.7
Tylvalosin	100	100	100	100	100	100	33.3	100	100
Vancomycin	0	0	0	0	0	0	0	0	0

Sensitivity = [number of true positives/(number of true positives + number of false negatives)] × 100%. Note: the number of tests performed to calculate detection sensitivity was 3.

**Table 6 antibiotics-11-00845-t006:** Sensitivity of overall protocol for 16 groups of antibiotics at spike-in concentrations of 10 μg, 5 μg, and 1 μg.

Antibiotics Group	Sensitivity								
	10 μg Spiked-in			5 μg Spiked-in			1 μg Spiked-in		
	Water Sample	Fecal Sample	Meat Sample	Water Sample	Fecal Sample	Meat Sample	Water Sample	Fecal Sample	Meat Sample
Aminoglycosides	33.3	33.3	40	20	13.3	33.3	0	0	0
Amphenicols	100	100	100	100	100	100	50	50	83.3
Antifolate	100	100	100	100	100	100	100	100	100
Carbapenems	0	0	0	0	0	0	0	0	0
Cephalosporins	40	40	40	40	40	40	33.3	33.3	20
Fluoroquinolones	60	60	60	60	60	60	60	60	60
Glycopeptides	0	0	0	0	0	0	0	0	0
Lincosamides	100	100	100	100	100	100	100	100	100
Macrolides	91.7	100	100	91.7	83.3	100	41.7	58.3	58.3
Nitroimidazole	100	100	100	100	100	100	100	33.3	66.7
Penicillins	66.7	66.7	66.7	66.7	44.4	66.7	22.2	0	11.1
Pleuromutilins	100	100	100	100	100	100	100	100	100
Polymyxins	0	0	0	0	0	0	0	0	0
Quinoxaline 1,4-di-N-oxides (QdNOs)	100	100	100	100	100	100	100	100	100
Sulfonamides	100	100	93.3	100	100	100	100	100	100
Tetracyclines	100	100	100	100	100	100	75	83.3	58.3

Sensitivity = [number of true positives/(number of true positives + number of false negatives)] × 100%.

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
