# Peer review of "A Universal LC-MS/MS Method for Simultaneous Detection of Antibiotic Residues in Animal and Environmental Samples"

_antibiotics, 2022, doi:10.3390/antibiotics11070845_

Round 1

Reviewer 1 Report

This is a well designed study about an important subject.

Reviewer 2 Report

In the paper A Universal LC-MS/MS Method for Simultaneous Detection of Antibiotic Residues in Animal and Environmental Samples, authors report demonstrate a methodology that permits simultaneous extraction and detection of antibiotic residues in animal and environmental samples. 

Paper is very interesting, expecialy by consisiderinf that multi screaning determination in LC MS is challange. 

Paper lack in several aspects. 

I suggest major revision

Specific comments

Page 2 Line 2

However, the common limitations of antibiotic residue testing in these studies are the lack of proven applicability of the testing protocol to detect different sample types and a wide number of antibiotic classes used in human and animals.

Pleas add that moreover this substances can be unstable. 

In this contex add references sucha as

Stability of Antibiotics, Pesticides and Drugs in Water by Using a Straightforward Procedure Applying HPLC-Mass Spectrometric Determination for Analytical Purposes. Separations, 2021, 8.10: 179.

Page 6 

3.1. Limit of Detection

Howe authors have obtain LOD? 

Please explain

Reviewer 3 Report

Tun et. al. have done a valiant attempt to develop a universal LC-MS/MS method for detecting broad class of antibiotics in animal and environment samples. The manuscript lacks on following comments and addressing them is necessary.

A.  Please mention total number of antibiotics and their classes in abstract. Discussion (Line153) and Conclusion (LIne234).

B. Please replace "tested" with "permits" in abstract as author's method could only detect ~70% antibiotics.

C. Format Line 14 in methods for clarity.

D. In section 2.2, please mention how antibiotics were added in solid sample like were samples mixed and whether samples were stored at a specific temperature and for how long before their chemical extraction.

E. It is not clear in section 2.3, how much is the methanol % of solvent used for chemical extraction. Is it 100% ? This is important to know because authors dilute the supernatant to 5% Methanol and this will be 20 times dilution.

F. In section 2.3, authors have extracted antibiotics from solid samples with 3 steps of 10 ml buffer. In Section 2.4, they dilute the supernatant to 5% Methanol. With these 2 steps, the antibiotics concentration will be markedly different for solid and liquid samples. Authors should provide these details to help readers know and adapt their method.

G. Please mention the length, thickness and catalogue numbers of MAX, HLB and MCX cartridges in section 2.4 to help readers adapt this method.

H. In section 2.4 and lines 67-68, please mention the ratio in which elutes were mixed from different cartridges.

I. Authors have arranged table 2 according to their retention time. Next tables in the manuscript are arranged alphabetically. It will be helpful to have Table 2 in alphabetical order as well.

J. Please mention units of transition 1 and 2 in Table 2.

K. Please explain how authors took Transition ranges for different antibiotics in Table 2. 

L. Please explain how LOD values were calculated in Table 2. DId authors inject different concentrations of antibiotics? Please show data for antibiotics with high LOD and low LOD as an example. In section 3.1, Lines 90-91, please clarify if LC-MS/MS was used or samples were infused directly in mass spectrometer.

M. Numbering of sections 3.1 onwards is wrong. Authors detected 30 antibiotics so authors should correct "12" to "13" in lines 99, 102.

N. In section 3.2, Solid phase extraction, it is not clear how authors calculated the % of recovery in Table 4. In the text, authors have accounted for 33 antibiotics but their results in Table 4, have 38 antibiotics detected.

O. In section 3.3, Lines 130-138, please explain how authors have calculated data for classes of antibiotics. Have they averaged the sensitivity of all antibiotics in a class or they have summed the data for all the tests done for all antibiotics in a class. This will help readers adapt this method.

P. In table 5, please mention the number of tests done to calculate detection sensitivity.

Q. In table 5, Ampicillin and Tiamulin have 100% sensitivity despite poor SPE recovery. This is in contrast to Amoxicillin, Cefquinome  and Meropenem which as expected have 0 sensitivity due to poor SPE recovery. Please explain.

R. Please mention SPE method instead of "our developed method" for clarity and not getting confused with LC-MS/MS.

Round 2

Reviewer 2 Report

All corrections were made. 

Paper can be accept

Reviewer 3 Report

Authors have satisfactorily answered all of my comments.